# The Transition to Adulthood in Children of Depressed Parents: Long-Term Follow-Up Data from the Family Talk Preventive Intervention Project

**DOI:** 10.3390/ijerph20043313

**Published:** 2023-02-14

**Authors:** Taylor L. Myers, Tracy R. G. Gladstone, William R. Beardslee

**Affiliations:** 1Wellesley Centers for Women, Wellesley College, 106 Central St., Wellesley, MA 02481, USA; 2Department of Behavioral and Social Sciences, School of Public Health, Brown University, 121 South Main St., Providence, RI 02903, USA; 3Boston Children’s Hospital, 300 Longwood Ave., Boston, MA 02115, USA

**Keywords:** parental depression, emerging adulthood, transition to adulthood, adolescence

## Abstract

Little is known about the effects of parental depression on offspring as they transition to adulthood—a challenging time developmentally, when late adolescents must separate from home, achieve intimate relationships, and develop a sense of identity. We present long-term quantitative and qualitative data from early adolescents with a depressed parent who were randomized to one of two family-based preventive interventions and followed over time, across the transition to young adulthood. Specifically, we present clinical measures of psychopathology and Likert-scale questionnaire data from young adults and their parents regarding the transition to adulthood and perceptions of the interventions. We also report in-depth qualitative interview data from young adults about the effects of parental depression on their transition to adulthood. Findings suggest that leaving home, establishing relationships, and coping with stressors may be challenging for emerging adults. Furthermore, the interviews highlight the importance of siblings, the burden of parental depression, and the development of self-understanding and empathy in young adults who grew up with a depressed parent. Data suggest that clinicians, policy makers, educators, and employers must address the preventive and clinical needs of young people and their families as they transition to young adulthood after growing up with depressed parents.

## 1. Introduction

Parental depression has been widely shown to have profound effects on children at each developmental stage [1]. For example, depressed mothers of newborns may be less attentive to their infants’ needs [2], and their infants often cry excessively [3,4] and have difficulty self-soothing [5] relative to offspring of non-depressed mothers. Maternal depression compromises bonding with the infant, which may lead to failure to thrive or attachment disorders [6]. Untreated postpartum depression can have effects on infants’ wellbeing into adolescence, including poor cognitive functioning, violent behavior, and externalizing disorders [7]. Further, young children of depressed mothers have been found to exhibit more fussiness and aggression relative to young children of healthy mothers [1,8]. Behavioral problems in young children, particularly boys, are also specifically tied to paternal depression beyond the effects seen from maternal depression [9,10].

During middle childhood and adolescence, relative to offspring of well mothers, children of mothers with recurrent or chronic depression have been judged to exhibit poorer social competence [11,12,13]. Preteen children of depressed mothers are more likely than preteens of non-ill mothers to have a negative interpretive bias for ambiguous events [14] and to blame themselves for negative events [15]. In addition, research on the connection between parental depression and adolescent functioning reveals that adolescents of depressed mothers have lower self-worth and self-esteem than do adolescents of well mothers [16].

Children of depressed parents are also more likely to develop depression themselves due to a number of factors including genetic predisposition and learned cognitions and behaviors [1]. In families with a depressed adult, disturbances are common in family functioning and parenting [17] including having fewer family routines [18] and more household chaos [19], which are associated with increased depressive symptoms among adolescent children of depressed parents. Within families with depressed parents, children of all ages are involved in caretaking [20,21]. Both instrumental caretaking (i.e., taking on additional responsibilities around the home) and emotional caretaking (i.e., comforting the depressed parent) are positively correlated with adolescents’ reports of anxiety and depressive symptoms [20]. Children of depressed parents may blame themselves for parental symptoms, which places them at further risk for internalizing symptoms themselves [22], and emerging adult children of depressed parents acknowledge the burden of navigating their parents’ moods [23,24].

While there is an abundance of research on young children and adolescents with depressed parents, there is a noteworthy dearth of research on the impact of parental depression during the life stage of emerging or young adulthood. Emerging adulthood (here defined as ages 18–25) is a relatively new life stage and is as yet understudied [25]. The transition from adolescence to emerging adulthood is often characterized by independence from guardians—typically through employment or higher education and leaving the home. Additionally, this life stage includes an emphasis on the formation of intimate relationship attachments [25]. When successful, these relationships, which may include cohabitation and deeper levels of intimacy, can be an important source of security and satisfaction for the young adult and serve as a valuable coping resource while negotiating other young adult challenges.

The expectation to complete the major life tasks associated with this life stage may precipitate or exacerbate depressive symptoms [26], and the unstable and “in-between” nature of emerging adulthood can have profound implications for mental health [25]. Given that mental healthcare providers are already struggling to offer sufficient services amidst a mental health crisis among youth, it is of utmost importance to ensure a successful transition to adulthood to promote mental wellness among young adults. Success in achieving the developmental tasks of emerging adulthood is made more likely by several factors including resilience [27], positive coping skills [28], perceived social support (including parental support) [29], sibling relationships [30,31], and having a family routine [18]. In contrast, factors such as childhood maltreatment [32], marital conflict between parents or divorce [33,34], and parent substance use [35,36] can make the transition to young adulthood more challenging. Despite knowing that depression in parents negatively impacts children and that adverse family contexts challenge the transition to adulthood, little is known about how young adults with depressed parents fare during the transition to adulthood.

In the following, we present longitudinal data on the course of psychopathology among depressed parents and their young adult children from a study of family-based preventive intervention programs. We also report perceptions of the transition to adulthood with a depressed parent through Likert-scale questionnaires with both young adults and parents and through qualitative interviews with young adults. Offspring of depressed adults are at higher risk for depression and ongoing experiences of stressors [1,37] and often feel responsible for parents’ symptoms [22], yet many children of depressed parents do not develop depression themselves [38,39]. Therefore, our specific hypotheses were as follows:

**H1.** *Emerging young adults will report elevated rates of depression themselves and will report having experienced multiple stressful life events*.

**H2.** *As they transition out of adolescence, these young adults will report a sense of guilt about parental depression and worry that leaving home will exacerbate parental symptoms*.

**H3.** *A significant number of these young people will report that they did well during the transition to adulthood despite these adversities*.

## 2. Materials and Methods

### 2.1. Study Design

A mixed method was used to conduct this study—semi-structured interviews were conducted to collect qualitative data, and questionnaires were employed to collect quantitative data. Participating families were randomly assigned to one of two interventions–a clinician-facilitated intervention (Family Talk) or a lecture-based intervention. Details of the randomized longitudinal study of the two family-based preventive interventions, including intervention allocation patterns and short-term intervention effects, have been reported previously [40]. 

### 2.2. Data Collection

Parents and children separately completed assessments at baseline (T1), following intervention (T2), and every 9–12 months after (T3–T8). In this mixed method, qualitative embedded within quantitative study, we report findings from (1) structured clinical interviews conducted at T8 with both young adults and their parents to assess current and past psychopathology; (2) ratings from Likert-scale questionnaires with both young adults and their parents about the transition to adulthood and perceptions of the project; and (3) qualitative interviews with young adults after T8 about the effects of parental depression on their transition to adulthood.

### 2.3. Study Sample

Participants were enrolled in a pilot study of a family-based depression prevention intervention between 1989 and 1992 (n = 20 families, 61 individuals), or in a NIMH-funded randomized controlled trial of the same intervention between 1992 and 1996 (n = 85 families, 267 individuals). At enrollment, families included (1) at least one child between the ages of 8 and 15 that was never treated for a depressive episode and (2) at least one parent that had experienced a depressive episode in the past 18 months. Detailed inclusion and exclusion criteria have been reported previously [40]. All families with young adult participants that turned eighteen during the course of the study were invited to participate in the assessments and interviews presented in these analyses. 

For the purposes of these analyses, participants from both the pilot and the NIMH-funded samples were combined to form the total sample of 105 families with 328 individuals. Through the sixth assessment point, 91 (87%) families with 273 individuals in the total sample were retained, and, at T8, 72 (69%) families representing 200 individuals remained in the study (Table 1). Of these, n = 114 parents and n = 83 children completed the clinical assessment at T8. 

Overall, n = 78 children and n = 74 parents completed the Likert-scale questionnaire—three parents completed the measure twice for a total of n = 77 parent interviews. The parents were assessed, on average, 9.8 years (SD = 1.8) after enrollment and ranged in age from 48 to 62 years (mean = 55.9, SD = 5.5) at this time point. The young adults were assessed on average 8.6 years (SD = 1.9) after enrollment. Table 2 shows the timing between baseline, T8, and the administration of the questionnaire.

A total of N = 50 young adults (mean age of 20.5 years, SD = 2.4) completed the qualitative interview (n = 29 in the lecture condition; n = 21 in the clinician-facilitated condition). The interviews were administered between T7 and T12. In thirty-one cases, the mothers had depression, in twelve cases, the father had depression, and in seven cases, both parents had depression. Thirty-six young adults were in school, fourteen were in the workforce and one was in the military. Forty-six of the fifty young adults had at least a high school education, and twenty-five of fifty were involved in community activities.

### 2.4. Measurement

#### 2.4.1. Measures of Parent and Young Adult Psychopathology

Parents were interviewed about current and past psychopathology using the Schedule of Affective Disorders and Schizophrenia-Lifetime version (SADS-L) [41], a semi-structured interview for diagnosing mood disorders and other psychopathology. Episodes in the interval between visits were assessed using the Streamlined Longitudinal Interval Continuation Evaluation (SLICE), an adaptation of the Longitudinal Interval Follow-up Evaluation [42]. The children and young adults were interviewed using the Kiddie-SADS-ER (K-SADS) [43], a modification of the SADS-L for child and adolescent populations. For the interval, a comparable child and adolescent adaptation of the SLICE (K-SLICE) was used [44]. Detailed descriptions of these instruments, their psychometric properties, and why they were chosen have been provided in previous publications [40].

#### 2.4.2. Likert-Scale Questionnaire Regarding Transition to Adulthood and Perceptions of Project

The questionnaire included seven-point Likert-scale ratings for young adults (Appendix A) and a comparable questionnaire for parents (Appendix B). Both the young adult and parent versions of the questionnaire collected information on the transition to adulthood, mental health treatment utilization for the young adult, and perceptions of the project. Questions in the parent version focus on late adolescence, about the time their child completed high school. Some mothers completed the questionnaire twice if they had more than one child in the study. The young adult questionnaire also asked about major life events (e.g., moves, school/job changes, troubles with close relationships, financial/legal difficulties, health concerns) since the last assessment time point. 

#### 2.4.3. Qualitative Interviews Regarding Transition to Adulthood

Young adult participants engaged in qualitative interviews addressing the following topics: general introduction and current occupational status, global functioning, major life events, coping with life events, experiences of parental illness, coping with parental illness, and feedback about the intervention. 

### 2.5. Analyses

The presence of affective or nonaffective illness was determined according to SADS, KSADS, SLICE, and K-SLICE criteria, which follow Research Diagnostic Criteria [45]. Diagnoses of major depressive disorder, bipolar disorder, and dysthymia were designated as affective illnesses, and other illnesses such as substance use disorder and psychosis were considered as nonaffective illnesses. Descriptive statistics were used for all Likert-scale items from the young adult and parent questionnaires. In analyzing the parental data, we chose to focus on one parent per family—the mother (unless maternal data was unavailable). Qualitative interviews were reviewed in terms of participants’ verbatim responses, and key themes were identified and summarized. 

## 3. Results

### 3.1. Quantitative Data Analysis Results

#### 3.1.1. Clinical Data on Parent and Young Adult Psychopathology

At baseline, when the unit of analysis was family-level illness, all families had parents with affective illness. In addition, 46 (44%) families had non-affective illness. Table 3 presents parent-level episodes of affective and nonaffective disorder using the parent as the unit of analysis.

Although our enrollment criteria included the absence of a history of affective illness in the children by parent report, we found upon direct assessment of the children at baseline that 19 of 138 (14%) children did in fact have a history of affective illness. Table 4 presents data on child-level episodes (Table 4a) and cumulative lifetime (Table 4b) affective and non-affective illness. Cumulatively, 52% of youths experienced episodes of mood disorder by T8. 

#### 3.1.2. Likert-Scale Questionnaire Responses

Descriptive data for all Likert-scale items from the young adult and parent questionnaires are shown in Table 5.

##### Leaving the Home

Despite the fact that most of these young adults were in their twenties, a large number still lived in the parental home. Eighteen (23.1%) of the participants had never lived away from the family home, although it is important to note that a number of these participants either were still in high school or had just graduated and were preparing to leave home for college at summer’s end. Of the remaining 60, 32 (41.0%) were currently living away from the family home, and 19 generally lived elsewhere when college classes were in session but were temporarily living at home during summer or winter break. Finally, nine (11.5%) of the participants lived out of the family home at some point during the interval, but were currently living at family home at the time they completed the questionnaire for some reason other than a break from college classes. Many participants moved in and out of the family home, and between apartments and dorms, during college and immediately afterward.

Most young adults felt the transition from living at home to independent living went reasonably well (mean = 5.6, SD = 1.4 on a 7-point scale, where 1 indicates it went ‘poorly’ and 7 indicates it went ‘very well’). In general, they found they were relatively ready to live away from home (Mean = 2.7, SD = 1.5). However, they also found the transition to be somewhat stressful (Mean = 5.1, SD = 1.6). 

Parents believed their youths did well in the transition to young adulthood (Mean = 5.4, SD = 1.8) but were not always sure that their youths were mature enough to be away from home (Mean = 2.9, SD = 1.7). The parents reported that they themselves experienced significant stress as their adolescents were making this transition (Mean = 3.8, SD = 1.4).

##### Experience of Stressful Events

All but one young adult provided responses to the Major Life Events (MLE) questions in the Likert-scale questionnaire (MLE; n = 77), and 76 participants reported one or more major life events occurring within the family during the interval in which the subject was assessed. Over half (51%) reported four or more MLE (see Table 6). Of the total sample, 58% of participants reported that they or a family member had moved during the interval, and 18% reported more than one move. Three-quarters of our young adult participants (75.3%) reported a transition between jobs and/or schools for themselves or a family member, and 21% reported more than one transition. One-quarter (25%) reported financial or legal difficulties. Forty-six percent reported either a significant medical problem (either their own or someone close to them) or death of someone close to them, and 13% reported more than one medical problem or death. Forty-four percent of participants reported other major life events.

Forty-seven out of fifty young adult respondents had a close relationship outside the family, and parents reported a considerable amount of concern over their children’s sexual activity (Mean = 3.4, SD = 1.9). However, 44% of young adults reported difficulties in their close relationships, including several noting increased conflict in their own romantic relationships. Romantic breakups, sometimes more than one, were common during the interval. Many participants reported drifting away from, or having a falling out with, certain friends. For those young adults who lived away from the family home at some point (n = 63), nearly one-third (32%) reported difficulties with their living situation.

##### Young Adults’ Experience with Depression

Thirty-one of forty-nine emerging adults felt they had suffered from depression or mania; however, some of the instances were mild and did not meet diagnostic criteria. For those emerging adults who had experienced their own mood disorders, they reported it was helpful to be able to talk about it with their parents (Mean = 4.7, SD = 1.8). Thirty out of forty respondents reported still talking to their parents about depression, yet only twenty reported being able to talk to parents about their own concerns for depression. Nine said they did not feel able to talk to their parents about their own risk for depression. 

Twenty-one of fifty emerging adults had used mental health services since leaving high school, and three received help through the original study. Those who used mental health services were often, but not always, satisfied with those services (Mean = 4.9, SD = 1.7). Young adults’ concern about availability of health care when they were no longer covered by their parents’ health insurance policy tended to be mild-to-moderate (Mean = 2.9, SD = 2.1). Parents also thought the health coverage their youths received was adequate (Mean = 5.4, SD = 1.7) but reported considerable concern about health insurance after the child was no longer covered by their health insurance (Mean = 5.4, SD = 1.7) and reported that access to health care was fairly difficult (Mean = 3.8, SD = 2.4). Most of the parents were satisfied when youths did get help (Mean = 4.5, SD = 1.8), and, when they sought help from the project, they felt quite satisfied with the resources they received (Mean = 3.8, SD =2.4). 

##### Perceptions of the Project

Overall, young adult participants were quite positive about the study, rating the question “Are you glad you are in the project?” (1 indicating ‘not at all happy’, 4 indicating ‘moderately happy’, and 7 indicating ‘very happy’), on average, 5.5 (SD = 1.3). Young adults felt the study helped communication with their parents moderately (Mean = 3.6, SD = 1.8), and noted that study participation helped improve their relationships with their parents somewhat (Mean = 2.6, SD = 1.7). Specifically, they rated ease of communication about family affective illness with their affectively ill parents, on average, 4.9 (SD = 1.6), and, only slightly higher (Mean = 5.0, SD = 1.4) with the non-ill parent. About half the respondents (16= yes, 17= no, and remainder either did not respond to the question or had no siblings) said they talked to their siblings about parental depression and found it helpful (Mean = 4.7, SD = 1.4, on a 7-point scale). According to the young adults, study participation also increased their understanding of affective illness considerably (Mean = 4.2, SD = 1.6). 

Young adults positively rated the study’s effect on seeking help for mental health concerns, rating the question, “Did [the project] make it easier or make you feel more comfortable about seeking mental health services?” (1 indicating ‘stayed the same’ and 7 indicating ‘made it much easier’), on average, at 4.5 (SD = 1.7). Parents agreed that the study made it easier for them to seek mental health services for either themselves or their children (Mean = 4.5, SD = 1.8). 

### 3.2. Qualitative Interview Responses

Representative quotes from the qualitative interviews, separated by theme and sub-theme, are presented in Table 7.

#### 3.2.1. Leaving the Home

The in-depth qualitative interviews with young adults revealed feelings of worry and guilt in response to moving away from the parental home with a depressed parent. One young man who joined the military said he felt he caused his mother to become depressed again by leaving and felt burdened by guilt. Another respondent reportedly increased her phone contact with her mother when she was away. A common theme among the young adults was feeling burdened by increased responsibilities in the home. A few said the depression made moving away from the parental home easier. 

The presence of siblings also had an effect on young adults’ feelings regarding leaving the home. Forty-one of the fifty families had siblings; half of these reported being emotionally close to their siblings, with their siblings providing a source of support. Older siblings reportedly worried about how their younger siblings would fare when left alone with the parents once they moved out or went away to college. 

#### 3.2.2. Establishing Intimate Relationships

In general, the presence of parental depression did not seem to have a negative impact on the quality of the parent/child bond. The majority of emerging adults felt they still had a close relationship with the depressed parent and often felt closer to the depressed rather than non-depressed parent. In only about one-third of the participants was any evidence found for a young person actively distancing themselves from the parental home. 

In terms of the impact of living in a family with depression on relationships, the young adult respondents most commonly reported that having a parent with depression made them more empathic (9). Another ten participants cited a negative impact, including that the embarrassment of having a depressed parent set them apart from their peers (3). Twelve of forty respondents felt it did not have an impact. 

Specifically considering peer relationships, fifteen young adults felt that growing up in a family with depression had been positive for peer relationships, most commonly resulting in greater compassion. Twelve felt the impact had been negative. Twelve of forty-two felt it had no impact, and the remainder of the responses did not fit a single category. In terms of the negative impacts, respondents reported they did not socialize as much because of their parents’ depression or worried that they might have exhibited the poor social skills of their parents. 

A recurring theme throughout the young adult interviews was the importance of siblings in coping with parental depression. Siblings spoke of the comfort of having someone else in the same situation. One young adult said that he had not even been able to confide his feelings to his closest friends, but he could talk to his siblings. In contrast to the benefits of siblings in coping, older siblings frequently found themselves in a parental role, filling in the gaps that disabled parents were unable to provide. The role of older siblings in providing guidance and fostering understanding of the parent’s depression was particularly apparent. 

#### 3.2.3. Experience of Stressful Events

As noted in the Likert-scale questionnaires, many of the young adults experienced one or more stressful life events in the six months prior to the interview. Several of the reported stressors may reflect the self-perpetuating effects of depression in the family, including a parent abusing alcohol, unemployed parents, and geographical moves due, in part, to marital strife or financial constraints. Many of the stressors are particularly characteristic of a family with one or more children transitioning to young adulthood. Many participants living on their own, for instance, and even a few starting their own families, reported that tight finances were stressful. A few found their substantial college loans to be overwhelming, and some struggled with bad credit. Many participants found the process of applying to colleges and making decisions difficult.

Some of the stressors reported did not seem necessarily unique to a family with depression, nor did they seem unique to a family in which a child was transitioning to young adulthood. Even so, these stressors may have precipitated or aggravated depression within the family. A few participants reported parents with medical problems, including a father having a hip replacement, another father with heart problems, and a mother with diabetes. A few participants reported stressful life events tied to the historical context in which these interviews took place—one was in the military and about to be deployed, two had brothers fighting in the war in Iraq, and another was very upset by the events of September 11th.

Particularly striking was the number of participants who reported one or more significant deaths. A large number of participants reported grandparents dying, and a few reported an aunt, an uncle, or close friend of their parent’s dying. Some reported having a close friend who lost a parent or sibling. One reported losing friends in an auto accident. Thankfully, none of these was an immediate family member, but the participants nonetheless had close connections to the deceased. Coping with death may not be unique to families with a young adult or families battling depression, but the nature of a young adult’s connection to the deceased raises an interesting issue regarding the dual effect of such a death, serving as a source of loss that also may impel young adults to fill supportive or caregiving roles.

#### 3.2.4. Perceptions of the Program

Forty of the fifty young people reported parental depression continued to be an important issue in the family. The qualitative interview sought to understand how participating in the study affected the emerging adults and their families. When asked the question, “Do you think being in the project made a difference in your family?” forty-seven participants responded. Thirty-six said yes, seven said no, and four were unsure. Participants reported, “It gave me some way to talk about my Dad’s depression”; and, “It opened up an opportunity to talk in the family and understand more.” Participants also cited improved awareness of the illness as being a positive outcome of the study. One young adult said, “It made everyone more aware of each other’s moods”; with another saying, “It helped me recognize their issues and my own issues with depression and helped me understand them better.” 

Three main themes emerged in response to the question about what might be helpful to other kids. These were: realizing it was not anyone’s fault, realizing that many families had to deal with depression so there was no need to feel ashamed, and recognizing the importance of understanding the situation and valuing family openness. 

One of our goals in the intervention was to educate children about the causes of depression and diminish any guilt, particularly in terms of the children taking responsibility for parental depression. Only five of the thirty-eight young adults who responded to this question felt they did not know what caused depression; one felt he had caused the illness and a second said raising children was a contributing factor. The majority of these young adults, twenty of thirty-eight who answered the question, felt that the source of the mood disorder was stressful life events. Many also blamed the parent’s difficult childhood, but marital strain and job dissatisfaction were also given as potential causes. Another eleven young adults felt it was a combination of stressful life events and genetic factors. Thirty of forty young adults stated unequivocally that they did not feel guilty or to blame for the parent’s depression. Another four said they felt some guilt earlier, but, as they grew older, had come to understand it was not their fault. Most of the rest gave responses that indicated they still had some struggle with the issue, at least to a certain degree. 

## 4. Discussion and Implications

The combined clinical and Likert-scale questionnaire data from the perspective of both young adults and their parents, in addition to qualitative responses from young adults, reveal insights into how parental depression may influence the transition to adulthood. Parents and their young adult children acknowledged that parental illness was an ongoing burden, as evidenced by 44% of parents experiencing affective episodes through T8. Although there are intervals in which some families reported no parental illness, it is remarkable how frequently affective illnesses recurred during the course of this study, regardless of whether the unit of analysis was the parent or the couple. Additionally, and in line with our first hypothesis, over half of the children also experienced affective illness over their lifetime, well above the prevalence rate of depression (17%) expected in the general population [46]. This finding is consistent with data from a number of longitudinal risk studies that found high rates of depression and other mood disorders in youths at risk for depression during the transition into young adulthood [47,48,49]. 

Most parents and their children reported that the transition to young adulthood went reasonably well, in support of our third hypothesis. The young adults felt the transition went better and they had been more ready to leave home than their parents. Still, both parents and young adults reported some stress during this transition. Although depression is associated with, and often precipitated by, major life events, it is nonetheless striking that the majority of participants had experienced three or more major events in the previous year. In particular, the frequency with which participants reported deaths of loved ones suggests that particular attention to grieving, loss, and moving on is an important task in young adulthood complicated by growing up with a parent with depression. 

Both young adults and their parents were satisfied with the mental health care services youth currently received, but parents were much more concerned than their children about their children’s health insurance once the young adult left the parents’ insurance plan (a rating of, on average, 5.4 for parents versus 2.9 for young adults). The concern demonstrated by parents is well-founded given that the transition to adult care for both physical and mental health is often managed poorly and results in discontinuation of services [50,51,52]. Given that these young adults are at increased risk for depression by way of their parents’ mental illness, it is crucial that they retain easy access to mental health care into adulthood. Both young adults and their parents positively rated study participation as making it easier or more comfortable to seek mental health care in the future. These reports echo the findings of Gulliver et al. (2010), who reported that past positive experiences with help-seeking, education, and awareness facilitated future help-seeking behavior [53], and encourage early intervention for families experiencing parental depression.

The qualitative data from the interviews with young adults did not vary by intervention condition. Both groups of young adults reported that families faced the continued presence of serious depression and other mood disorders in parents and that a number of young adults wrestled with depression themselves. It is clear that these young adults worry not only about their parents and leaving them, supporting our second hypothesis, but also about themselves. It is also clear that, for many, the presence of a parental mental illness heightened their awareness and compassion for that parent and for suffering in general. Many of these young adults report that their experience with a depressed parent led them to develop significant interpersonal empathy and awareness of the inner worlds of their parents, their friends, and their siblings. Interestingly, these reports contrast with the findings of Apter-Levy et al. (2013) in which young children of depressed mothers displayed decreased empathy compared to children of non-depressed mothers [54]. Likewise, Pratt et al. (2017) found that parental depression is associated with decreased empathy in offspring by identifying its neural basis in preadolescents of depressed mothers compared with preadolescents of non-depressed mothers [55]. It appears that, to date, the connection between parental depression and empathy in children of depressed parents has not been explored in young adults, but data from this investigation suggest the possibility that, as children enter adulthood, the experience of parental depression may possibly enhance their warmth and compassion for others. 

In understanding the resilience of young people, a notable finding is the large role played by siblings in the context of parental depression. Many older siblings reported that they served as stand-in caregivers for their younger siblings. Others reported feeling guilt or concern over leaving their younger siblings in the home as they moved out or attended college. In a meta-synthesis on the experience of adult children of parents with mental illness, Murphy et al. (2011) also noted this theme of parentification, with older siblings having to take care of younger siblings [56]. In addition, young adult participants cited siblings as a source of companionship and shared experience. Like sibling relationships in the wake of divorce [57], this may have been protective for these youths as they managed the challenges of growing up with a depressed parent. 

Given the significant number of young adults who, passing through an increased period of risk, developed affective disorders during this study, there are important public health implications for the mental health of emerging adults. Programs directed at at-risk offspring, whether they are primarily prevention, education, or treatment programs, must provide ways for young people to obtain treatment, when needed, quickly, adequately, and without barriers. Additionally, parents and young adults strongly endorsed the value of our two preventive intervention strategies even up to ten years after intervention, supporting the notion that preventive interventions delivered at one developmental epoch can have sustained effects. Taken together with our finding that parental depression continues to affect offspring into emerging adulthood, this information points to the delivery of preventive intervention strategies that provide families with skills that can be used over the long term (i.e., self-understanding, building relationships, being active, and accomplishing tasks), and should support parents in helping their children to acquire these skills.

## 5. Conclusions

The findings presented here must be considered in light of several limitations. First, the data were collected from the early 2000s, nearly twenty years ago, and as such cannot provide insight into how more contemporary concerns—such as social media use [58,59], climate anxiety [60,61], and social and economic injustice [62,63]—influence the mental health, identity formation, and transition to adulthood in emerging adult children of depressed parents. Second, this study lacks qualitative interview data from the parents of our young adults, limiting our ability to understand parents’ perspectives during this transition. Third, interrater reliability ratings are unavailable for the qualitative interviews. Lastly, approximately half of the young adults in this study participated in a longitudinal family-based intervention about parental depression that, as has been shown here, had positive effects on the parent-child relationship. Although young adults in the lecture-based intervention did not have direct involvement in that intervention, they participated in intensive assessment throughout the study and were asked to reflect on their experiences growing up with a parent with depression. As such, the experiences of these young adults during the transition to adulthood may not be generalizable to young adults with depressed parents who were not engaged in a study about parental depression. 

These limitations notwithstanding, the data presented here speak to themes of family, adversity, and developmental trajectories that persist across generations and highlight the importance of attending to the effects of parental depression on offspring’s transition to adulthood in future work. Specifically, more research is needed to better understand the transactions between depressed parents and children when they are no longer living in the same home. In addition, emerging adults are living at home longer at the time of writing compared to at the time of data collection [64]. As such, future work must evaluate the processes of establishing independence as an emerging adult while living at home with a depressed parent. Our findings also call attention to the need for preventive intervention approaches for families with parental depression to address the long-term effects of this chronic illness. Our finding that sibling relationships were protective against the effects of parental depression suggests that interventions aimed at enhancing the quality of the sibling relationship should be explored. 

While parental depression remained a chronic issue, and depression in the young adult children emerged as a new problem in many cases here, many of these children did well. The dimensions of self-understanding, empathy, compassion, and relatedness that many participants exhibited suggest areas to target in future preventive intervention efforts focused on building resiliency in families with parental depression. Through adolescence, offspring of depressed parents have higher rates of depression, other diagnoses, major life events, and impairment in social functioning. As such, it is crucial that we consider the effects of parental depression on youth as they leave adolescence and enter the adult world.

## Figures and Tables

**Table 1 ijerph-20-03313-t001:** Total enrollment and demographics by assessment point.

	Baseline	T2	T3	T4	T5	T6	T7	T8
Families, n	105	97	95	96	92	91	86	72
Persons, n	328	304	298	292	286	273	247	200
Parents, n (Clinician, Lecture)	190 (106, 84)	177 (100, 77)	173 (97, 76)	170 (98, 72)	165 (94, 71)	159 (91, 68)	141 (80, 61)	115 (60, 55)
Parent age mean (min, max)	43 (30, 56)	44 (31, 56)	44 (32, 57)	46 (33, 58)	47 (34, 59)	47 (35, 59)	49 (36, 60)	50 (37, 61)
Children, n (Clinician, Lecture)	138 (78, 60)	127 (73, 54)	125 (71, 54)	122 (69, 53)	121 (69, 52)	114 (64, 50)	106 (58, 48)	85 (41, 44)
Child age mean (min, max)	12 (9, 15)	12 (9, 16)	13 (9, 17)	14 (10, 18)	15 (11, 19)	16 (12, 20)	17 (13, 22)	18 (14, 25)

**Table 2 ijerph-20-03313-t002:** Time to semi-structured interview.

	Baseline to T8 Assessment	Baseline to Questionnaire	T8 to Questionnaire
Parents; N, mean years, SD (min, max)	74, 6.8, 1.3 (4.7, 10.2)	77, 9.8, 1.8 (6.7, 13.8)	74, 3.0, 1.9 (−0.1, 6.4)
Children; N, mean years, SD (min, max)	67, 6.7, 1.4 (4.6, 10.4)	78, 8.6, 1.9 (4.8, 13.8)	67, 2.1, 2.0 (−0.8, 6.4)

**Table 3 ijerph-20-03313-t003:** Parent-level illness.

	Baseline	T3	T4	T5	T6	T7	T8
Total parents, N	190	175	170	164	157	141	114
n (%) affective illness							
Total	135 (71)	112 (64)	92 (54)	81 (49)	71 (45)	66 (47)	50 (44)
Clinician	73 (69)	62 (63)	54 (55)	49 (53)	40 (44)	40 (50)	31 (53)
Lecture	62 (74)	50 (66)	38 (53)	32 (45)	31 (46)	26 (43)	19 (34)
n (%) nonaffective illness							
Total	52 (27)	45 (26)	41 (24)	39 (24)	31 (20)	31 (22)	23 (20)
Clinician	33 (31)	25 (25)	26 (27)	23 (25)	17 (19)	21 (26)	11 (19)
Lecture	19 (23)	20 (26)	15 (21)	16 (23)	14 (21)	10 (16)	12 (21)

**Table 4 ijerph-20-03313-t004:** (**a**). Child-level illness; (**b**). Child cumulative lifetime illness (N = 138).

**(a)**							
	**Baseline**	**T3**	**T4**	**T5**	**T6**	**T7**	**T8**
Total Children, N	138	125	122	121	113	102	83
n (%) affective illness							
Total	19 (14)	28 (22)	26 (21)	26 (21)	34 (30)	28 (27)	20 (24)
Clinician	9 (12)	12 (17)	14 (20)	15 (22)	17 (27)	15 (28)	10 (17)
Lecture	10 (17)	16 (30)	12 (23)	11 (21)	17 (34)	13 (27)	9 (20)
n (%) nonaffective illness							
Total	54 (39)	53 (42)	51 (42)	44 (36)	39 (35)	29 (28)	22 (27)
Clinician	24 (31)	26 (37)	26 (38)	22 (32)	21 (33)	16 (30)	11 (28)
Lecture	30 (50)	27 (50)	25 (47)	22 (42)	18 (36)	13 (27)	11 (25)
**(b)**							
	**Baseline**	**T3**	**T4**	**T5**	**T6**	**T7**	**T8**
Affective illness							
%Total	14	27	33	40	46	51	52
%Clinician	12	23	29	67	42	47	49
%Lecture	17	32	38	43	52	55	57
Nonaffective illness							
%Total	39	44	48	49	50	52	53
%Clinician	31	37	42	44	46	49	50
%Lecture	50	53	55	55	55	57	57

**Table 5 ijerph-20-03313-t005:** Likert-scale Ratings.

	Question	N	Mean	Std Dev	Min	Max
Young Adult Responses	1. Transition out of home	40	5.6	1.4	2.0	7.0
	2. Readiness to live away	39	2.7	1.5	1.0	6.0
	3. Stress	37	5.1	1.6	1.0	7.0
	4. Total life events	19	4.7	2.8	1.0	11.0
	5. Mental health services	21	4.9	1.7	1.0	7.0
	6. Quality of care	46	2.9	2.1	1.0	7.0
	7. Was project helpful	47	4.5	1.7	1.0	7.0
	8. Helpful to talk with parent	51	4.9	1.6	1.0	7.0
	9. Helpful to talk with other parent	31	5.0	1.4	1.0	7.0
	10. Talk to siblings	26	4.7	1.4	1.0	7.0
	11. Own illness	46	4.7	1.8	1.0	7.0
	12. Project helped talking	74	3.6	1.8	1.0	7.0
	13. Relationship change	72	2.6	1.7	1.0	7.0
	14. Increased understanding	74	4.2	1.6	1.0	7.0
	15. Glad to be in project	74	5.5	1.3	1.0	7.0
Parent Responses	1. Transition to higher education	74	5.4	1.8	1.0	7.0
	2. Level of maturity	77	2.9	1.7	1.0	6.0
	3. Sexual activity	77	3.4	1.4	1.0	7.0
	4. Substance abuse	77	3.0	1.9	1.0	7.0
	5. Overall stress	77	3.8	1.4	1.0	7.0
	6. Adequacy of health coverage	77	5.4	1.7	1.0	7.0
	7. Coverage after leaving family	59	3.8	2.0	0.0	7.0
	8. Difficulty in accessing health care	75	3.8	2.4	0.0	7.0
	9. Satisfaction with help received	31	4.5	1.8	0.0	7.0
	10. Satisfaction with project	17	5.1	2.0	1.0	7.0
	11. Help from project	63	5.1	1.8	1.0	7.0

**Table 6 ijerph-20-03313-t006:** Major Life Events ^1^.

	Move	Job/ School Changes	Trouble with Close Relationships	Difficulties with Living Situation ^2^	Financial/ Legal Difficulties	Death or Medical Problems	Other	Total
Mean	0.857	1.052	0.545	0.373	0.260	0.714	0.558	4.273
Standard Deviation	0.942	0.857	0.717	0.613	0.470	1.145	0.819	2.905
Min, Max	0.4	0.3	0.3	0.3	0.2	0.7	0.5	0.13
Number of events (%) 0 1 2 3 >3	41.6 40.3 10.4 6.5 1.3	24.7 54.5 11.7 9.1 0.0	55.8 36.4 5.2 2.6 0.0	67.8 28.8 1.7 1.7 0.0	75.3 23.4 1.3 0.0 0.0	54.5 32.5 7.8 1.3 3.9	55.8 37.7 3.9 1.3 1.3	1.3 14.3 15.6 18.2 50.6

^1^ One participant was also given the young adult questionnaire but provided no MLE responses so was not included in the N. ^2^ n = 59, figures represent only participants who lived out of the family home at some point during the study.

**Table 7 ijerph-20-03313-t007:** Direct quotes from qualitative interviews with participating young adults relating to the themes addressed in the interviews.

Main Themes	Sub Themes	Quotes	Condition ^1^
Leaving Home	Abandoning depressed parent	*“I feel guilty leaving home. I felt like I was leaving them behind, afraid things wouldn’t be the same.”*	C
	Abandoning siblings	*“Our parents don’t understand being a teenager now. They are not social. They don’t adjust to new friends … My sister is starting high school. I’d have liked to be around for that.”*	C
Establishing Intimate Relationships	Negative impacts	*“I didn’t invite friends to my house. My parents are generally not in a good mood.”*	L
		*“I find it too hard to fit in with other kids…I didn’t do well with peers.”*	L
		*“I wonder if I wasn’t as trusting of other people as I might have been. My dad had trouble letting go…neither of my parents are extroverted…I don’t socialize much either…I have trouble making friends.”*	C
		*“I’m not sure if depression had anything to do with the fact that I didn’t make a lot of friends in high school. I wasn’t disliked, but I didn’t initiate social things. I waited for things to happen.”*	C
	Siblings aiding in coping	*“It’s helpful; we are all going through the same thing.”*	L
		*“Sharing a common problem made us closer.”*	L
		*“When Mom is in a bad mood and she blows up at us, we look at each other and know what is going on. We are not alone.”*	C
	Parentification of older siblings	*“[My older brother] only had to give us a stern look when mom wasn’t feeling well. We were well behaved.”*	C
		*“I told them she was sad. I told them she was sick. It wasn’t [their] fault. She will get better when she takes medications.”*	C
		*“I do [talk] in a mediator role. I explain her behavior in the context of depression.”*	L
		*“He (older brother) understands me. He puts things in a better framework. He understands it.”*	L
Perceptions of the Program	Enhanced communication	*“It gave me some way to talk about my Dad’s depression.”*	L
		*“It opened up an opportunity to talk in the family and understand more.”*	L
	Enhanced understanding of depression	*“It made everyone more aware of each other’s moods”*	C
		*“It helped me recognize their issues and my own issues with depression and helped me understand them better.”*	C
		*“When we were younger, I felt I did something like having a messy room. As we got older, that seemed irrational. We learned it wasn’t our fault.”*	C
		*“At first, I thought it was my fault; it felt like a burden. Then I realized it was part of depression.”*	L
		*“There is the typical, ‘It’s not your fault’ thing. When I was younger, I knew she was sad a lot, but didn’t connect it. At the same time, there was this little voice inside my head which said ‘Maybe it is your fault.’”*	L
	Benefits to other youth	*“Learning to take care of one’s self, realize what has happened, and then to do the best you can.”*	C
		*“Understanding that it’s a fairly common problem; it doesn’t make you or your family weird…don’t blame yourself; it’s not your fault.”*	L
		*“I learned to adapt, not to spend much time at home and to go to other houses. I learned to be patient; things will eventually get better.”*	L

^1^ C = Clinical-Facilitated Intervention; L = Lecture-Based Intervention.

## Data Availability

The data presented in this study are available on request from the corresponding author.

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
