# Peer review of "The Transition to Adulthood in Children of Depressed Parents: Long-Term Follow-Up Data from the Family Talk Preventive Intervention Project"

_ijerph, 2023, doi:10.3390/ijerph20043313_

Round 1

Reviewer 1 Report

Dear authors, I like your research topic and have some questions and recommendations. Please consider them. 

What do you mean by "Likert-scale interviews"? Scale and interview are completely different from each other. 

What is your research method? I think your method is mixed methods. If so, what's your mixed method type? Please explain it. 

What's your sampling method type?

"SADS, KSADS, SLICE, and K-SLICE criteria." What do you mean?

Your findings are clear and understandable. 

The discussion section is a lot bit weak. You should discuss your results with current literature studies. 

I could not see your recommendations. It should be added. 

Reviewer 2 Report

I find this manuscript well written. Although the authors pointed it out as a limitation that the data used were from decades ago, I think it would be good if the authors can provide more context why they used this dataset, and how the findings could be still relevant in the current times. Also, the authors mentioned that the participants were assigned to two conditions, but I find differences in results between the conditions were not much described. 

Reviewer 3 Report

The article describes an interesting topic. However, there are a number of problems with methodology and explanation of context, as well as some gaps in the study’s results description, that recommend this article be revised before it should be considered for publication. The rest of these comments are oriented toward formative feedback to help the authors to improve the text.

Introduction:

It looks very good, but the last paragraph I it needs to be reformulated and arranged, especially with regard to mentioning the hypotheses here.

There is no literature review, which provides the theoretical support for this paper. The article needs to establish the theoretical ground of the study.  Therefore, it is better to add a new section for the theoretical framework and hypotheses, in addition to that the hypotheses must be supported by previous studies. In addition, it is better to write the hypotheses as follows:

H1:

H2:

H3:

Materials and Methods:

I feel ambiguous about the methodology.

Study Design:

At the beginning of the Study Design section, the authors can clearly state the methodology (qual, quan, or mixed) and research model/design (e.g survey, interview, or explanatory sequential mixed design, etc).

2.2.1 Measures of Parent and Young Adult Psychopathology:

Before this sub-section, I suggest that a section be added under the name (Measurement), in which describe the methodology used in this paper, and clarified and describe the tools used in the study (questionnaire and interview).

2.2.1 Measures of Parent and Young Adult Psychopathology

Line 125-126 “using the Schedule of Affective Disorders and Schizophrenia-Lifetime version (SADS-L). Give more information about this Schedule and its items or attach this Schedule as Appendix.

2.2.1 Measures of Parent and Young Adult Psychopathology  and 2.2.2 Semi-Structured Interviews Regarding Transition to Adulthood and Perceptions of 132 Project  were as a questionnaires  not Interview.

2.2.2 Semi-Structured Interviews Regarding Transition to Adulthood and Perceptions of 132 Project

The description of this measure is not clear. The authors called this measure as semi-structured interview but they described it as a questionnaires or survey.

2.3 Analyses:

In line 150, SADS, KSADS, SLICE, and K-SLICE criteria. The authors need to clarify these criteria, what does they mean?

-The authors need to add a new sub-section about (Data Collection).

Materials and Methods:

I feel ambiguous about the methodology.

Study Design:

At the beginning of the Study Design section, the authors can clearly state the methodology (qual, quan, or mixed) and research model/design (e.g survey, interview, or explanatory sequential mixed design, etc).

2.2.1 Measures of Parent and Young Adult Psychopathology:

Before this sub-section, I suggest that a section be added under the name (Measurement), in which describe the methodology used in this paper, and clarified and describe the tools used in the study (questionnaire and interview).

2.2.1 Measures of Parent and Young Adult Psychopathology

Line 125-126 “using the Schedule of Affective Disorders and Schizophrenia-Lifetime version (SADS-L). Give more information about this Schedule and its items or attach this Schedule as Appendix.

2.2.1 Measures of Parent and Young Adult Psychopathology  and 2.2.2 Semi-Structured Interviews Regarding Transition to Adulthood and Perceptions of 132 Project  were as a questionnaires  not Interview.

2.2.2 Semi-Structured Interviews Regarding Transition to Adulthood and Perceptions of 132 Project

The description of this measure is not clear. The authors called this measure as semi-structured interview but they described it as a questionnaires or survey.

2.3 Analyses:

In line 150, SADS, KSADS, SLICE, and K-SLICE criteria. The authors need to clarify these criteria, what does they mean?

-The authors need to add a new sub-section about (Data Collection).

Results:

3.1 Participants:

This sub-section is best integrated into the Materials and Methods section under the Study Sample.

- Result: This section needs to be divided in two. Firstly, the “Quantitative data analysis results” section. Secondly, the “Qualitative data analysis results” section.

- 3.4 Qualitative Interview Responses  in Line 288 P. 12

The qualitative data analysis results, which were taken into account when analyzing the interview transcripts could be shown in a table using examples. So, the results (main themes and sub-themes) must be presented in tables. In addition, it is necessary to include the level of agreement between the researchers on a sample of comments, chosen at random.

Discussion :

in line 423 The authors need to add the and Implications with the  Discussion section to be Discussion and Implications , and add some paragraphs about Implications. In addition, the paragraphs about the Limitations need to be in the Conclusions, Limitations and Future Work section

Conclusions:

in line 520 The authors need to add the Limitations and Future Work to be Conclusions,

Limitations and Future Work , and add some paragraphs about Future Work.

I hope that these comments, oriented toward formative feedback, will help the authors to improve the text. Good job.

Round 2

Reviewer 3 Report

The researchers did an excellent job. Now, I recommend accepting this paper after making some very simple modifications, which are as follows:

-Under the study design part: the researchers should add one sentence at the beginning of the first paragraph (Mixed  method was used to conduct this study, semi-structured interviews were conducted to collect qualitative data, and  questionnaires were employed to collect quantitative data.)

-In Table 7, I suggest that another column be added after the Theme column, with the title of Sub-Themes, and the first column with the title of Main Themes. For example, Leaving Home (Main theme) and the sub-themes that fall under it: 1- Abandoning depressed parent 2- Abandoning siblings.

Best regards,

Author Response

Point 1: Under the study design part: the researchers should add one sentence at the beginning of the first paragraph (Mixed  method was used to conduct this study, semi-structured interviews were conducted to collect qualitative data, and  questionnaires were employed to collect quantitative data.)

Response 1: Thank you for this suggestion to clarify the methodology. We have added this sentence to better describe the study desing.

Point 2: In Table 7, I suggest that another column be added after the Theme column, with the title of Sub-Themes, and the first column with the title of Main Themes. For example, Leaving Home (Main theme) and the sub-themes that fall under it: 1- Abandoning depressed parent 2- Abandoning siblings.

Response 2: We appreciate this suggestion for formatting the table. We have added the Sub Themes column to better illustrate the content of the table.